# Big Data in Gastroenterology Research

**DOI:** 10.3390/ijms24032458

**Published:** 2023-01-27

**Authors:** Madeline Alizadeh, Natalia Sampaio Moura, Alyssa Schledwitz, Seema A. Patil, Jacques Ravel, Jean-Pierre Raufman

**Affiliations:** 1The Institute for Genome Sciences, University of Maryland School of Medicine, Baltimore, MD 21201, USA; 2Department of Medicine, Division of Gastroenterology and Hepatology, University of Maryland School of Medicine, Baltimore, MD 21201, USA; 3Veterans Affairs Maryland Healthcare System, Baltimore, MD 21201, USA; 4Marlene and Stewart Greenebaum Cancer Center, University of Maryland Medical Center, Baltimore, MD 21201, USA; 5Department of Biochemistry and Molecular Biology, University of Maryland School of Medicine, Baltimore, MD 21201, USA

**Keywords:** gut microbiome, genomics, metabolomics, proteomics, transcriptomics, epigenomics, medical informatics

## Abstract

Studying individual data types in isolation provides only limited and incomplete answers to complex biological questions and particularly falls short in revealing sufficient mechanistic and kinetic details. In contrast, multi-omics approaches to studying health and disease permit the generation and integration of multiple data types on a much larger scale, offering a comprehensive picture of biological and disease processes. Gastroenterology and hepatobiliary research are particularly well-suited to such analyses, given the unique position of the luminal gastrointestinal (GI) tract at the nexus between the gut (mucosa and luminal contents), brain, immune and endocrine systems, and GI microbiome. The generation of ‘big data’ from multi-omic, multi-site studies can enhance investigations into the connections between these organ systems and organisms and more broadly and accurately appraise the effects of dietary, pharmacological, and other therapeutic interventions. In this review, we describe a variety of useful omics approaches and how they can be integrated to provide a holistic depiction of the human and microbial genetic and proteomic changes underlying physiological and pathophysiological phenomena. We highlight the potential pitfalls and alternatives to help avoid the common errors in study design, execution, and analysis. We focus on the application, integration, and analysis of big data in gastroenterology and hepatobiliary research.

## 1. Introduction

Unquestionably, the availability of massive datasets, so-called ‘big data’, has revolutionized biomedical research. While the contents of large datasets vary greatly, their defining characteristic is size, containing anywhere from hundreds of thousands to millions of entries. As described herein, handling such large, diverse, high-throughput datasets requires a set of analytical and biostatistical skills that have developed as separate disciplines in concert with the growth and complexity of such datasets. Understanding the utility, applications, and drawbacks of each type of dataset is crucial to their proper analysis and integration, and necessary to avoid pitfalls in their handling, storage, and interpretation. 

In this comprehensive review, we focus on understanding how the use of big data has become an indispensable feature of gastroenterology and hepatobiliary research. These fields are particularly well-suited to the use of big data because of their unique positioning at the nexus between the gut (mucosa and luminal contents), the brain (enteric nervous system), the immune and endocrine (enteroendocrine cells) systems, and the vast GI microbiome. We believe such an analysis is timely given the emergence of novel, more powerful, less costly tools that provide exquisite genetic and epigenetic and proteomic detail at the cellular and subcellular level. Of course, as investigators learn to navigate this complex technological landscape, the potential for misapplication, misinterpretation, and misuse of such techniques, particularly by those unsophisticated or untutored in their practice, is a concern. Thus, our goal is to review the use and analysis of common types of research datasets comprising genetic, genomic, epigenetic, epigenomic, transcriptomic, proteomic, and metabolomic information. In this context, we also address specific issues concerning the analysis of the gut microbiome and the use of the clinical data derived from electronic health records and medical imaging. We describe and illustrate a representative study design that integrates these features in the context of inflammatory bowel disease (IBD) research. 

Although the reader is encouraged to delve more deeply into these issues, perhaps using the citations as a starting point, we discuss the relevant aspects of the data distribution, preferred analytical approaches, common pitfalls, and advanced statistical methods that, in many cases, were developed specifically to address the complexity and integration of big data. To avoid the common pitfalls and mistakes in the use of such complex information requires the standardization of the analytical and statistical methods and the expanded training of a cadre of bioinformaticians adept in their use; this is clearly team science.

## 2. Common Large and High-Throughput Datasets 

Multiple types of large datasets exist and are used within the realm of GI and hepatobiliary research. An overview of the most common data types is discussed below, and the prominent benefits and limitations to their use are addressed in Table 1.

### 2.1. Human Genetics and Genomics 

The traditionally heritable component of GI diseases can be explored in many ways, which are largely characterized as genetic or genomic approaches. Genetic approaches focus on single genes, whereas genomic approaches take a broader view to study the interplay between large swaths of the genome. Commonly, genetic studies focus on monogenic contributions to disease, involving in vitro [16,17] and/or animal models [18]. These approaches are used to characterize rare diseases, such as monogenic autoinflammatory disorders [19], as well as common diseases with a monogenic etiology, such as Maturity Onset Diabetes of the Young (MODY) and familial hypercholesterolemia [20]. The impact of this research extends beyond individual diseases. For example, the identification and characterization of the *MEFV* gene and its role in Familial Mediterranean Fever (an autoinflammatory disorder) elucidated the roles of the inflammasome and dysregulated innate immunity in health and disease [21]. 

Genome Wide Association Studies (GWAS) enroll tens to hundreds of thousands of participants, followed by wide-scale genomic analysis to seek the variations in the genome associated with outcomes or diseases in a predictive (as opposed to causative) manner [1]. Several bioinformatic and statistical steps are employed to assess these variations, including prioritization of the single nucleotide polymorphisms (SNPs) that may be related to the process under consideration [22] and corrections for multiple comparisons [23,24]. The additional steps required to pool data for meta-analysis [25] offer the ability to study specific SNPs in more diverse contexts [26]. GWAS have been conducted to seek the links between SNPs and atherosclerotic disease [27], the monocyte–leukocyte ratio (often used as a predictor of the response to infectious disease) [28], and even suicidal behaviors [29,30]. GWAS are also useful in characterizing unknown mechanisms by identifying which genes are playing a role in a given process or condition [31,32] and, in this context, can even be used to identify drug targets [33]. 

### 2.2. Epigenetics and Epigenomics 

Epigenetic modifications are non-heritable changes caused by covalent “tags” or other modifications of the genome. Similar to genetics and genomics, the fields of epigenetics and epigenomics examine specific and broader patterns of epigenome changes, respectively [34]. Epigenetic tags include methylation, which commonly suppresses gene expression, and acetylation, which decreases histone binding, thus freeing portions of the genome to increase expression; epigenetic tags modulate cell type-specific patterns of gene regulation and expression. CpG islands, or regions with multiple CG repeats, are common sites of gene methylation, often guiding development and cell differentiation; heavily condensed regions are referred to as heterochromatin. A well-described developmental example is X chromosome inactivation [35,36]. More changes occur in somatic cells than initially believed, and various exposures and diseases can also induce changes in these patterns [36,37]. Histone acetylation often correlates with demethylation; the resulting open or free DNA is referred to as euchromatin [35].

Both epigenetics and epigenomics are studied using a variety of techniques which take advantage of covalent modifications to characterize methylation and acetylation patterns. These include sequencing-based methods, such as bisulfite sequencing, ChIP-seq, and ATAC-seq [38]. The role of epigenetics and epigenomics in understanding disease causation and identifying potential therapeutic targets has been explored in cancer [39,40], aging [41], and weight regulation and metabolism [42,43], as first demonstrated by the Dutch famine study [44]. Understanding the role of epigenetic modifications in weight gain and weight-related disorders (such as metabolic syndrome) is of particular importance to the fields of endocrinology and metabolism, as well as GI and hepatology; over the past 20 years, obesity-induced liver steatosis, fibrosis, cirrhosis, and cancer have reached alarming proportions.

### 2.3. Transcriptomics 

Transcriptomics describes the study of RNA expression, which includes microRNAs (mRNAs), non-coding RNAs (small and long ncRNAs), interfering RNAs, and small nuclear and nucleolar RNAs (sn/snoRNAs), amongst others. RNA-seq or RNA sequencing can be used to sequence specific subsets of RNA; while previously popular array-based assays targeted expression analysis, most high-throughput methods are RNA-seq-based (e.g., mRNA-seq, total RNA-seq, targeted RNA-seq, single cell (sc) and single nucleus (sn) RNA-seq, and small RNA-seq) and can capture all expression data in a sample, permitting the exploration of new and unknown RNAs [45,46]. These approaches can be used to answer different questions and are unique in their strengths and weaknesses: for example, unlike scRNA-seq, snRNA-seq can be applied to frozen samples; this avoids dissociation bias, but it is limited to providing information about nuclear transcripts [47]. 

Bulk or total RNA-seq is the most common transcriptome sequencing method and can cost-effectively offer great insight into physiological processes. The technique is performed on a homogenized sample of tissue, serum, or some other sort, and sequencing is performed on the entirety of the sample in a non-selective, unbiased manner. This technique is often used to compare physiologic differences under varying conditions, such as exposures and treatments, disorders and diseases, and in the context of genetic variation. However, the existence of multiple cell types often complicates analysis; the additional use of cell-type markers in analysis as a means of identifying cell-type abundance changes has been used to estimate such changes [48]. However, non-marker changes cannot be attributed to a specific cell type and thus require further mechanistic characterization to identify cellular involvement. 

Bulk RNA-seq, the oldest and least costly RNA-seq approach, is quickly being supplanted by more sophisticated approaches. scRNA-seq provides cell-specific gene expression data. Many investigators have now shifted to the scRNA-seq-based interrogation of transcriptomes, but cost may be limiting—reduced sample size requirements for accurate analysis may help offset these costs. Furthermore, the access to machines for data generation is somewhat limited, and despite the requirement for fresh samples, the options for preservation media may be inadequate. In scRNA-seq, individual cells from a tissue sample are separated and, after sequencing, reads are mapped to a reference transcriptome to estimate the RNA expression levels; following normalization, the expression levels of marker genes are analyzed for each cell type, though this process can be thorny [49]. Multiple attribution tools and the increasing use of machine learning-based methods can help to optimize cell type assignments [50,51]. Common concerns include the limited identification of rarer cell types [52,53], the impact of high dimensionality on unsupervised clustering [54], and the mixed expression patterns displayed by some cells (i.e., multiple responses to the same signal) [55]. A lack of adequate quality control at every step can make the results untrustworthy [56,57]. Furthermore, the inappropriate input and use of classifiers can adversely affect the downstream results and interpretation [58,59]. Examples of scRNA-seq versatility are the finding of new neuronal subtypes [60] and hormone-producing cell types [61] and the characterizing of sex-specific transcription patterns at the cellular level [62]. Relevant to GI research, this approach has identified the axial spondylarthritis immune cell changes specific to IBD [63].

### 2.4. Proteomics 

Large-scale, protein-level proteomics can complement DNA- and RNA-level studies or be used independently to seek the potentially targetable mechanisms underlying disease. The many steps between the translation and the production of proteins include folding, signal sequence cleavage, functional group additions such as glycosylation, and protein–protein binding to form complexes; this makes it difficult to predict the biological consequences of transcriptional changes alone without protein-level confirmation. Nonetheless, it is important to consider the potential reasons for the discordance between the RNA and protein expression patterns [64]. Instead of protein microchips with limited identification abilities [65], chromatography and mass spectrometry methods are often used to explore protein presence and abundance [66]. Protein folding and interactions can be assessed using chemical crosslinking, affinity purification, protein correlation profiling, and proximity labeling among others [66]; this characterization permits the recognition of protein networks, with pathway- and context-dependent functional annotation [67] that may identify disease biomarkers and drug targets. The popularity of proteomics studies has been especially evident in oncology, where biomarkers with high specificity and sensitivity are difficult to identify [68]; an example is the recategorization and subtyping of breast cancer, which impacts treatment stratification [69,70,71,72]. 

### 2.5. Microbiome 

The microbiome represents a collection of microbial (primarily bacterial) species that inhabit the human body in a largely tissue-specific manner elucidated by the Human Microbiome Project [73]. The individual taxa or genus can be assessed using 16S rRNA amplicon sequencing or shotgun metagenomics; while 16S amplicon sequencing amplifies variable regions of the 16S gene to assess taxa abundance, metagenomic sequencing relies on Nextgen Sequencing, and read assignments are used to assess taxa abundance and genetic functionality. The intestinal microbiome has a deeply intertwined and dynamic relationship with several GI disorders. Other sites of microbial colonization include the oral cavity, skin, nasopharynx, and vagina, which have site-specific characteristics. A “healthy” vaginal microenvironment is characterized by *Lactobacillus* predominance with one of four *Lactobacillus* species [74], whereas the colon microbiome is considered to be ‘healthier’ when more diverse [75,76,77]. 

Defining a “normal” microbiome can be difficult [8] and depends on the technique used to assess composition [7]; for example, swabs or stool samples may yield varying results when compared to tissue biopsy [9], and metagenomic sequencing approximates compositionality differently than 16S rRNA amplicon sequencing [78]. Yet, many studies have successfully identified and distinguished differences in the microbiome in human disease [79]; the intestinal microbiome in particular has been associated with many diseases, including arthritis, IBD, autism, and cancer, to name only a few [9,80,81,82]. Some causative and mechanistic studies have been performed, demonstrating the physiologic changes that occur in response to the introduction of certain bacteria; however, these studies are largely limited to animal models, which differ in microbiota composition and the potential systemic impact in comparison to humans [7]. Much work in humans is correlative rather than mechanistic [83]; to define the precise role of the microbiome in health and disease, longitudinal, interventional, and mechanistic studies are needed.

### 2.6. Metabolomics 

Metabolomics, or the high-throughput study of metabolites present in a sample, is an important potential addition to any multi-omic study of GI health and disease. Metabolites include both endogenous and exogenous compounds with molecular mass < 1500 kDa [84]. Similar to proteomics, metabolic profiles offer a deeper glimpse into the mechanistic underpinnings of biological processes and can be used to study the impact of genome and epigenome, microbiome, transcriptome, and even proteome differences and compositional changes for low molecular weight molecules [85]. Given its large size and the uncountable number of different compound classes, measuring the metabolome requires a range of mass spectrometry methods, including LC-MS, GC-MS, CE-MS, and IMS-MS, as well as NMR, which is less sensitive but provides nondestructive sample measurements [85,86]. Commonly, third-party companies generate the data used by researchers; this may complicate the analysis and interpretation since the raw data are usually not released and missing values are more difficult to interpret [87]. This is a particular concern for investigative teams who lack the training and knowledge to interpret and critically analyze such outsourced data. 

Metabolomics studies have provided useful insights into rheumatoid arthritis [88], ischemic heart disease [89], infectious diseases [90], and cancer [91]. Metabolomics research has also been used to explore the dynamic relationship between host and microbiome metabolism [92], an ideal approach to gaining mechanistic insights into digestive diseases.

### 2.7. Medical Informatics 

Medical informatics describes the study of patient clinical data, including laboratory and other clinical results, using a variety of statistical methods. Modeling is an umbrella term used to describe some of these processes, including various forms of regression [93,94,95], whereby a mathematical description (i.e., an equation) is derived to predict relationships among a set of variables, in this case clinical variables. For example, modeling clinical and demographic data can be used to predict outcomes. Depending on the outcome of interest and the format and type of data available, different methodologies and techniques can be used to predict associations. While smaller datasets can be used for these analyses, the robustness and accuracy of the model, as well as its predictive power, tend to increase with more input data. Thus, large datasets are especially useful in finding trends that are distinguished by small but potentially meaningful differences. Large datasets of clinical and demographic data also allow for the use of multiple predictor variables in tandem [96], identifying temporal associations in clinical outcomes [97], and applying machine learning tools to enhance predictive power [98]. In recent years, such datasets have become more accessible as a result, primarily, of medical institutions mandating the use of integrated electronic health records (EHR) [99] and the development of large national and other databases geared towards collecting extensive human data [100]. However, EHR do not generally provide a comprehensive assessment and are, therefore, susceptible to a broad range of biases, which must be kept in mind when analyzing these data. Nevertheless, in tandem with enhanced biostatistical approaches, the exploitation of these extensive datasets can enhance confidence in the generalizability and potential applicability of the resulting findings [101], which are crucial to achieving meaningful clinical impact.

### 2.8. Imaging Data 

Diagnostic imaging, using multiple modalities, is widespread and provides another data-rich resource. As with any tool requiring human interpretation, there is potential for error; efforts are underway to increase diagnostic accuracy using machine learning and artificial intelligence to correctly identify anomalies [102,103], reduce noise, and enhance the visualization of data prone to motion sensitivity [104]. Bioinformatic tools have been developed for these purposes [105,106,107] and will be key for the widespread use and incorporation of large bioimaging datasets. 

## 3. Big Data in Gastroenterology and Hepatobiliary Research 

GI disorders, influenced by genetic and environmental factors, are difficult to treat because of their long duration, recurrence, and persistence. Outstanding questions remain within the field, such as: what are the genetic underpinnings of different GI pathologies, such as colorectal cancer (CRC)? How do our internal and external environments impact their development? How can we use omics studies to understand these diseases and identify novel biomarkers to inform clinical decision making? Such questions need to be addressed in a multi-faceted approach. The use of high-throughput omics stands at the breakthrough of such discoveries (Figure 1). In this section, we examine how big data derived from various omics techniques are applied to gain a more comprehensive understanding and to advance the field of gastroenterology. We explore the application of human genetics and genomics, epigenomics, transcriptomics, proteomics, microbiome, and metabolomics to characterize multifactorial diseases such as IBD, to advance biomarker discovery, to generate novel ways to detect GI cancers and their progression earlier, and to develop precision medicine. Furthermore, we examine examples of how different technologies within medical informatics can integrate big data for useful analysis, and how imaging data can revolutionize procedures by assisting with the diagnosis and outcome predictions. 

### 3.1. Human Genetics and Genomics 

Genetic and genomic studies can enhance the mechanistic understanding of GI and hepatobiliary diseases, provide novel diagnostic tools, and pave the way to targeted interventions. Gastroenterologists generally obtain a basic understanding of the genetics of GI disorders at the start of their training from textbooks and lectures that focus on, among other topics, the role of genetic mismatch repair defects in Lynch syndrome [108], the importance of *KRAS* and *BRAF* mutations in colorectal cancer [109], and *HFE* testing for hereditary hemochromatosis. However, the impact of genetics in the GI field is not restricted to these diseases. Whole exome sequencing and GWAS can identify possible causal genes for other, sometimes undefined, syndromes and can help to elucidate their pathogenesis. For example, genomic studies associated the *MEFV* mutation with recurrent abdominal pain and fever in FMF, a rare monogenic disease [110]. A more common condition, celiac disease, has considerable genetic influence; compared to 30% of the general population, 90-100% of patients possess either class II HLA-DQ2 or -DQ8 [111,112]. Serology testing for tissue transglutaminase-immunoglobulin A (TG2-IgA) remains a crucial part of celiac disease diagnosis, but HLA typing has value in the excluding of celiac disease in seronegative persons [111].

GWAS advanced the understanding of the susceptibility alleles in multifactorial diseases such as IBD and Behcet’s disease. Over a decade ago, Jostins et al. reported more than 163 genetic loci associated with susceptibility to the two major forms of IBD, ulcerative colitis (UC) and Crohn’s disease (CD) [113]; since then, at least an additional 80 IBD-associated genetic loci have been reported. Of these, approximately two-thirds are shared with other complex diseases or traits, including primary immunodeficiencies and mycobacterial disease. As an example, *NOD2*, the first susceptibility gene for CD, is also associated with Blau syndrome, an autoinflammatory disease [114,115], highlighting the observation that genomics can the unravel the underlying mechanisms between seemingly distinct pathologies. Behcet’s disease, in turn, is an autoinflammatory disease characterized by painful mucocutaneous ulceration that can be associated with intestinal inflammation [116,117], and can even mimic CD. 

Genomics also paved the way for genotype–phenotype correlations, improved diagnostic testing and disease screening, and targeted treatments. Recently, the International Society for Gastrointestinal Hereditary Tumors (InSiGHT) applied a standardized classification scheme to a database containing variants of the *MLH1*, *MSH2*, *MSH6*, and *PMS2* genes associated with Lynch Syndrome and tied this information to clinical recommendations [108]. Furthermore, after Samadder et al. reported increased heritable variants in solid cancers by multigene vs. targeted testing [118] and Uson et al. reported that universal multigene panel testing increased the detection of the heritable germline mutations associated with CRC (15.5% harbored 62 pathogenic variants with *MSH2* among the most common genes) [119], the National Comprehensive Cancer Network recommended germline multigene panel testing for all individuals younger than 50 years with CRC. This highlights the contribution of genetics and genomics to understanding disease etiology, recognizing susceptibility alleles, and enhancing screening and early detection.

### 3.2. Epigenomics 

Epigenetic modifications such as DNA methylation are characteristically stable, making them a useful sensor of disease risk and progression. Their impact in the pathogenesis of diseases of the GI tract, such as colorectal and gastric cancer and IBD, highlight their complexity. Epigenomics can also be leveraged for the early detection of colorectal adenomas and cancers. In GI neoplasia, tumor suppressor genes may be inactivated by mutation or promoter and/or CpG island hypermethylation. Hypermethylated genes can be involved in cell cycle regulation, DNA repair, growth regulation, apoptosis, cell attachment, and signal transduction. *MLH1*, a mismatch repair gene, is an important example. In sporadic colorectal and gastric cancer, altered *MLH1* leads to microsatellite instability and can confer resistance to chemotherapeutics [120,121]. Arnold et al. demonstrated that the demethylation of hMLH1 in hypermethylated cell lines led to the re-expression of the protein and reversed resistance to 5- fluorouracil [121]. Less commonly, mutation-negative families with suspected Lynch syndrome due to silenced *MLH1* expression in tumors may harbor constitutional epimutation, wherein hypermethylation at the promoter of one allele silences its expression in major somatic tissues [122]. Aberrant patterns of histone modification play a role in CRC by leading to transcriptional changes. In gastric cancer, the genes involved in cancer-related pathways are more frequently affected by epigenetic rather than genetic alterations [120].

Genome-wide DNA methylation profiling revealed changes in early neoplasia that can be leveraged as biomarkers using blood and feces [123]. Because they account for tumor heterogeneity, biomarker panels derived from genome-wide studies may be more advantageous than single markers. Two biomarker panels with high sensitivity and specificity in the detection of CRC are reported. One panel screens for promoter hypermethylation of *CNRIP1*, *FBN1*, *INA*, *MAL*, *SNCA*, and *SPG20*. For CRC, the combined sensitivity and specificity of at least two positives among these six genes are 94% and 98%, respectively [124]. The second panel discovered that *CDO1, DCLK1, SFRP1*, and *ZSCAN18* were frequently methylated in 71–92% of colorectal, gastric, and pancreatic cancers [125] with a combined sensitivity and specificity of 95% and 98%, respectively. 

Altered DNA methylation in IBD identified potential intestinal-specific inflammation biomarkers and predictors of colitis-associated CRC [126,127]. Current research on the DNA methylome in IBD distinguished the methylation changes due to environmental cues from those dependent on genetic susceptibility. Agliata et al. discovered that IBD intestinal methylome abnormalities are related to upstream genetic variants [128], suggesting an effector role for DNA methylation. Differentially methylated positions were also enriched in inflammation-related pathways downstream of cytokine signaling. This suggests that DNA methylation enhances the response to anti- and pro-inflammatory signals and that the IBD methylome comprises combined environmental and genetic cues [128]. Altered methylation can be used as a biomarker to predict the risk of CRC prior to the presence of dysplasia in colonic biopsies of high-risk IBD patients. When performing a genome-wide DNA methylation analysis via Illumina Human Methylation 450K BeadChip in biopsies from high-risk vs. low-risk patients, the prevalence of methylation was higher in the high-risk IBD patient population. To complement the current standard of care, a DNA methylation signature focused on five genes (*SLIT2*, *EYA4*, *FLI1*, *SND1*, *USP44*) may enhance the predictive value of surveillance biopsies for colitis-associated colon cancer [129]. This highlights how epigenomic studies may unravel GI disease pathogenesis and impact clinical practice.

### 3.3. Transcriptomics 

In recent years, the advent of diverse transcriptomics methods such as scRNA-seq has enabled enormous advances in GI and liver research. One area that has flourished in the wake of the evolving omics techniques is the study of GI and liver cancers. Transcriptomics was vital to uncovering four consensus molecular subtypes (CMS) of CRC. This is perhaps the most theoretically sound classification of CRC to date and its development was highly dependent on genomic, epigenomic, transcriptomic, and proteomic analyses [130]. Using RNA-seq in conjunction with available CMS data, others identified correlations between these CRC subtypes and the presence of bacterial species (identified by 16S rRNA amplicon sequencing) [131]. Similarly, molecular subtypes were identified and validated for gastric cancer, a malignancy known for high levels of heterogeneity even within the same patient [132,133]. Clinical applications to exploit these recent advances are under investigation, with investigators using transcriptomic methods, particularly scRNA-seq, to identify markers of prognosis and therapeutic response [4,5,131].

scRNA-seq profiling of colon epithelial cells permitted the identification of previously unknown epithelial cell subtypes relevant to inflammation, as well as potential therapeutic targets. Parikh et al. compared colon epithelial cells from immunotherapy-naïve IBD patients and healthy controls. Among other findings, scRNA-seq identified spatially segregated cells in different stages of differentiation and revealed a novel absorptive colonocyte that regulates luminal pH. These investigators also observed differences in goblet cell location and transcriptional activity between samples from UC patients compared to control normal colon tissue; additional types of inflammation-associated goblet cells were found in colon tissue from subjects with IBD. Notably, WFDC2, an anti-protease expressed by goblet cells, was downregulated in those with active UC, suggesting that it plays a role in maintaining mucosal barrier integrity [134]. Whereas there has long been support for the “leaky gut” hypothesis of IBD (i.e., the idea that the primary insult of an inappropriately permeable intestinal epithelial barrier introduces luminal antigens to the innate immune system and a cascading series of events leads to intestinal autoimmunity and inflammation), this study identified a potential therapeutic target in the pathogenesis leading to an inflamed intestinal barrier [135]. Others implemented combined GWAS and scRNA-seq to identify potential risk genes and infer their functions based on cell type-specific transcriptomes in UC [136,137]. Similarly, a meta-analysis comparing non-alcoholic fatty liver disease and alcoholic liver disease (NAFLD and ALD) was able to identify a subset of similarities in differentially regulated pathways, such as the regulation of fibrosis via *COL1A1* and *COL3A1*, which may be responsible for their similar and highly overlapping clinical phenotype [138]. Such approaches leveraging both genomic and transcriptomic methods may be particularly useful in the study of polygenic diseases such as IBD and fatty liver disease (NAFLD and ALD), which have many clinical phenotypes.

Perhaps the most revolutionary aspect of transcriptomics’ growing presence in GI research is its potential for personalized or precision medicine. As the barriers to affordability and implementation of RNA-seq are addressed, we can begin to uncover the extent of the pathophysiological diversity in GI and hepatobiliary disorders and discover new therapeutic targets.

### 3.4. Proteomics 

Because the protein makeup of a system is so closely linked to its physiologic function, the clinical importance of proteomics is evident as we progress toward personalized medicine. The use of proteomics technologies has led to great advances in GI research, especially in IBD [139]. One great challenge in the treatment of IBD in the era of biologic drugs is finding the most suitable therapy for a particular patient, while avoiding the adverse events related to medications and the overtreatment of patients whose disease may be milder. The application of proteomics to the predicting and monitoring treatment response in IBD is new, and most of the proposed predictive markers for more commonly used biologics are of limited clinical value [139,140]. Generally, the monitoring of patient response to a newly initiated therapy is accomplished by assessing trends in inflammatory markers in serum, such as C-reactive protein, and in stool, such as fecal calprotectin [141]. Although the trending of objective biomarkers alongside patient symptoms has been shown to be superior to clinical management alone in achieving the timely escalation of care to help IBD patients achieve mucosal healing, there is a need for more specific indicators of treatment response [142]. An ongoing, large multicenter prospective cohort study may shed light on this topic [143]. Pierre et al. used a shotgun approach to identify candidate proteomic biomarkers for CD relapse either less or more than 6 months after discontinuing infliximab, as short-term relapse occurs by a different mechanism than mid/long-term relapse [144]. The panels of 15 and 17 proteins, respectively, had higher predictive capability of disease relapse than C-reactive protein or fecal calprotectin. As more predictors of disease activity and response to specific drugs are identified by proteomics methods, information like this could be applied in the clinical setting to aid in the decision making regarding the choice of treatment and its continuation or withdrawal. Because of its high-throughput capability, proteomics is also likely to be useful in drug discovery and mechanistic research, as evidenced by a recent study on potential drug targets for infliximab-resistant UC, the first reported use of proteomics in drug discovery [145].

In treating CD, it is important to be mindful of the disease phenotype, including the presence of perianal disease, and the structuring or penetrating behavior, as this can impact the disease progression and decision making related to medical treatment and surgical intervention [146]. Several recent groundbreaking studies have shown that the application of proteomics may help to uncover the mechanisms underlying the presentation of these disease phenotypes and also to predict which CD patients are at risk of developing related complications. In a pilot study by Townsend et al., eight serum protein biomarkers were identified which could differentiate the CD patients who had a surgically resected intestinal stricture from the CD patients without stricturing behavior, with an area under the curve (AUC) > 0.9; additional biomarkers distinguished them from patients with UC. Those proteins differentiating stricturing from non-stricturing CD included those involved in complement activation, fibrinolytic pathways, and lymphocyte adhesion, although correlating the proteins with a mechanism of stricturing was beyond the scope of the study [147]. In a multicenter prospective observational study, Wu et al. implemented both proteomic analysis and the conventional enzyme-linked immunosorbent assay (ELISA) to identify the biomarkers that may help predict whether a pediatric patient with the inflammatory phenotype of CD will convert to a stricturing phenotype within three years. Of the 10 candidate serum biomarkers, 4 were significantly different in abundance at the time of CD diagnosis in patients who would convert to stricturing behavior within three years versus those who would not: extracellular matrix protein 1 (ECM1), cartilage oligomeric matrix protein, matrix metalloproteinase 9, and fibronectin. High levels of ECM1 (>3900 ng/mL) at the time of diagnosis proved to be the strongest predictor of conversion to the stricturing phenotype, with an accuracy of 75.4% and a sensitivity and specificity of 80.0% and 70.7%, respectively [148]. The rapid evolution of technologies such as machine learning also benefits the application of big data techniques in GI. In another prospective cohort of pediatric CD patients, Ungaro et al. used ensemble machine learning methodology to identify a panel of serum protein biomarkers measured at diagnosis which together could predict the time to the development of a stricturing or penetrating phenotype of CD with greater reliability than serologic studies [149]. 

Although endoscopy with histologic evaluation of mucosal biopsy samples provides the gold standard for diagnosing IBD, non-invasive proteomic biomarkers may become useful in the classification and stratification of IBD, and potentially even diagnosis. Many studies have examined differences in protein expression profiles in the intestinal tissue and serum of patients with IBD vs. healthy controls [150,151] and of UC vs. CD [147,150]. Starr et al. studied IBD in a pediatric population, where the traditional diagnostic features are more difficult to discern than in adult patients and found that a panel of five proteins could be used to differentiate active IBD from healthy colon tissue samples with an AUC of 1.0. In the same study, a panel of 12 proteins could differentiate pediatric patients with UC from those with CD with an AUC of 0.95 in the discovery cohort; in the validation cohort of patients, the panel accurately classified 80% of patients [152]. The additional validation of proteome findings is required before they can be used as diagnostic biomarkers or to distinguish between IBD and other potential causes of symptoms that may be mistakenly attributed to IBD, particularly for those with IBD in remission. Cytomegalovirus (CMV)-induced colitis is an example of such a potential confounder. In the future, these methods may become clinically useful when it is difficult to differentiate between IBD and another form of colitis or when endoscopy is contraindicated.

In addition to IBD, proteomics may be applied to other areas of GI research. Many studies have identified various panels of proteins relevant to GI cancer progression. Li et al. examined tissue samples of precancerous gastric lesions and gastric cancer, creating a model that could predict the progression of a gastric lesion to cancer with an AUC of 0.88, outperforming the prediction based on clinical risk factors alone [153]. Research is ongoing to identify the appropriate biomarkers in irritable bowel syndrome (IBS), as the vast range of clinical presentations and the overlap with other functional disorders presents challenges to the diagnosis, treatment, and outcome monitoring of patients with IBS [154]. Some studies reported differences in the serum or intestinal proteome composition of normal controls vs. IBS patients with diarrhea, constipation, or alternating diarrhea and constipation as the predominant symptoms; these studies were limited by small sample sizes [155,156].

### 3.5. Microbiome 

The gut microbiome represents the trillions of symbiotic bacteria, archaea, viruses, and eukaryotes, as well as their genes, residing in the GI tract that interact with the host to impact human physiology and disease [7]. The mechanisms whereby the microbiome influences host physiology is a topic of ongoing investigation, but likely includes activation of the innate immune system by microbial antigens and epigenetic modifications of host cell DNA and histones [157,158]. Because there is great inter- and intrapersonal diversity in gut microbiome composition, and it changes throughout life, a useful definition for the “healthy gut microbiome” remains elusive [7,158,159]. The key to understanding the scope of the microbiome’s role in GI physiology lies in the use of big data, which complements the culture-based methods. In monozygotic twins of concordant body size (i.e., body mass index > 30 kg/m^2^ or 18.5-24.9 kg/m^2^) and shared childhood environment, fecal 16s rRNA sequencing identified a common set of gut microbial *genes* shared between the twins rather than a common set of species [160].

Dysbiosis is one of many factors implicated in the etiology of IBD, but a direct causal relationship has not yet been elucidated. Proinflammatory gut microbes are associated with increased mucosal permeability, possibly through the release of toxins or the disruption of the pre-existing microbial community, thereby activating the innate immune system and inflammation [161]. Individuals with IBD have a less diverse gut microbiome compared to unaffected but otherwise matched controls, although there are conflicting reports regarding which species demonstrate altered abundance [9,161]. There is evidence that those with IBD have a lower abundance of Firmicutes in both stool and mucosal samples, which is an interesting finding given the phylum’s ability to metabolize dietary fiber to produce anti-inflammatory short-chain fatty acids [9,161,162,163]. Fewer studies found a higher abundance of mucolytic bacteria in IBD, including some *Ruminococcus* species, although this has been contested by others [164,165]. The roles of non-bacterial microbes in the gut should not be overlooked. The overgrowth of *Candida albicans* and other fungal species has been observed in IBD, particularly during a symptomatic flare; this is consistent with the proposed mechanism that *C. albicans* causes a Th17-mediated immune response during a disease flare [166,167]. Gut bacteriophages appear to drive both innate and adaptive immunocyte expansion through IFN-γ and Toll-like receptor signaling, and bacteriophages isolated from the feces of patients with active UC are more adept at inducing the CD4+ T cell production of IFN-γ compared to those from persons with inactive UC or healthy controls, suggesting the complicity of phages [168].

The gut microbiome has been implicated in the development of diverticular disease, but the mechanism remains unclear. Metagenomic studies attempted to profile the differences in resident microbiota in diverticular disease, but the results vary widely regarding which taxa, if any, are over- or under-represented in patients (reviewed by Ticinesi et al. [169]) [170,171,172,173]. With the added use of ^1^H nuclear magnetic resonance to examine metabolomes, Barbara et al. identified potential microbiota-related biomarkers relevant to diverticular disease, including higher urinary concentrations of kynureine, quinolate, and certain carbohydrates in individuals with asymptomatic diverticular disease versus controls, and lower urinary concentrations of hippurate in symptomatic diverticular disease [172]. Although probiotics are commonly prescribed to treat diverticular disease, there are insufficient data to support their use [174,175,176].

The liver also interacts with the gut microbiome as it receives microbiota-derived nutrients and toxins via hepatic circulation and regulates the innate immune response (reviewed by Wang et al. [177]). In an international, multi-center observational study, the bacterial compositions of the gut microbiomes of patients with alcohol-associated hepatitis differed from those of the healthy controls, with decreased diversity at the genus level based on the 16s rRNA sequencing of fecal samples. The investigators also found disproportionately low levels of *Akkermansia* and high abundances of *Veillonella*, both of which correlated with the model for end-stage liver disease (MELD) scores [178]. Other studies found increased relative abundances of *Veillonella* species in alcohol-associated cirrhosis compared to other causes of cirrhosis and in cirrhotics with hepatic encephalopathy (HE) versus those without HE [179,180]. Species of the genus *Akkermansia*, which support intestinal mucosal integrity and reduce hepatic fatty acid synthesis, were depleted in persons with alcohol use disorder even *without* cirrhosis and their quantity negatively correlated with the serum levels of pro-inflammatory cytokines and chemokines [181,182]. In mouse models, the ethanol consumption-induced depletion of *Akkermansia* was prevented by oral *Akkermansia* supplementation, leading to reductions in serum alanine transaminase, hepatic IL-1 β, and hepatic neutrophil infiltration compared to the non-supplemented mice with the same ethanol intake [183]. These studies suggest that gut microbiota play an early role in the pathogenesis of alcohol-associated liver disease (ALD) and that these microbes might be harnessed for therapeutic purposes.

One of the most well-known examples of the use of microbial supplementation as therapy is that of fecal microbiota transplant (FMT) for recurrent pseudomembranous colitis caused by *Clostridioides difficile* infection (CDI). In FMT, microbes are donated by transplanting the stool of a healthy donor into the intestinal tract of a recipient. There are multiple known mechanisms by which the donor microbiota restore eubiosis, including promoting the bacterial conversion of primary bile acids into secondary bile acids and re-introducing the bacterial species that produce anti-inflammatory short-chain fatty acids (reviewed by Soveral et al. [184]) [185,186]. The first case report of FMT for the treatment of CDI was published in 1983, although historical documents describe the use of FMT as early as the 4th century C.E.; with the U.S. Food and Drug Administration’s first approval of an FMT product in November 2022, we will likely experience an increase in the clinical use of and research regarding FMT [187,188]. FMT is being studied for potential application in other GI illnesses, including IBD, IBS, nonalcoholic steatohepatitis, and hepatic encephalopathy, as well as several metabolic, autoimmune, autoinflammatory, neurodegenerative, and psychiatric diseases [189,190]

### 3.6. Metabolomics 

Metabolomic studies can provide insights into the effects of diet and the environment on GI diseases. The imbalance of intestinal microorganisms due to GI disease can lead to inflammation and the consequent metabolic disturbances, primarily mediated by intestinal microbial metabolites such as bile acids, short chain fatty acids (SCFAs), and amino acids. Metabolomics integrates these diverse signals to elucidate pathway interactions, allowing metabolites to serve as molecular readouts of cell status [191]. To understand the differences and changes in the metabolome of host cells and intestinal microorganisms and to inform potential biomarkers, metabolic profiling can be applied to an array of diseases such as IBD, IBS, gastritis, celiac disease, and fatty liver disease (NAFLD and ALD). Metabolomics can also be leveraged to understand why those with IBD and IBS continue to experience visceral hypersensitivity despite disease remission. A shift in the microbiome in IBD results in the predominance of Gram-negative organisms and a decrease in colon microbiome diversity. In mice, compared to the controls, elevated SCFA levels were detected after the induction and recovery of colitis [192]. Thus, metabolic profiling can provide insights into these diseases. In individuals with IBD vs. the healthy controls, studies have identified aberrant metabolites associated with amino acids and bile acids, creatinine, alpha-glucose, and membrane components that can serve as biomarkers in plasma, colon biopsies, or feces [193,194]. There is less research on IBS metabolomic profiles, but some studies identified differences in volatile organic compounds in fecal samples and increased esters in diarrhea-predominant IBS [195,196]. Differences in the fecal profiles of those with ALD, including decreased SCFA production due to changes in microbiome composition [197], also reveal possible disease targets.

Visceral hypersensitivity is characterized by chronic pain in the absence of inflammation. Microbial metabolites can mimic neurotransmitters by binding nociceptors in neurons, thereby modulating visceral hypersensitivity [192,198]; microbial metabolites can directly stimulate nociceptors via TRP channels [199] and increased transient receptor potential vanilloid-1 receptor (TRPV1) expression in the rectum correlates with the severity of abdominal pain [192,200]. After incubation with cultured naïve dorsal root ganglion neurons, increased SCFAs increase TRPV1 sensitization, suggesting a pro-nociceptive role for microbial metabolites [192]. Leveraging metabolomics to study disease pathogenesis and identify potential biomarkers in diseases of the GI system is likely to enhance the mechanistic understanding of these disorders. A study of the effects of exercise in NAFLD took this a step further and examined the metabolomic profiles of adipose tissue, plasma, urine, and fecal samples, allowing for tissue-specific pattern identification. Babu et al. found that adipose tissue profile shifts correlated most closely with changes in the clinical parameters, offering an example of the type of finding necessary for therapeutic targeting [201].

### 3.7. Medical Informatics 

Medical informatics utilizes the intricacies of biomedical data derived from multi-omic studies and electronic health records (EHRs) to answer specific scientific questions relevant to the gastroenterology field. The tools within medical informatics, which include data modeling, text mining and natural language processing, and machine learning [202], can be used to create disease-based cohorts, draw associations, and conduct quality-of-care standard assessments. The EHR may contain extremely useful patient information meant to provide well-rounded patient care, allow communication between physicians, and serve a data reservoir. It comprises structured data elements, such as patient demographics and lab values, interpreted by common data models, and unstructured data elements, such as free text or narratives [203]. Through text mining and natural language processing, the unstructured data can be computerized and interpreted. The use of natural language processing has the potential to increase the accuracy of EHR case definitions in IBD, for example, because it integrates the disease information (i.e., pathologic findings) buried in narrative reports rather than solely relying on billing codes for diagnostic purposes [204]. This allows for the more efficient development of disease cohorts and opportunities for translational research [204,205]. Furthermore, machine learning approaches can be used in EHRs and omics data to unravel the underlying factors in multifactorial diseases. It has been used to predict genetic markers in IBD through GWAS [206], to classify CD patients based on their genetic signatures [207], or to integrate multi-omics data to unravel the factors that mediate intestinal dysbiosis in IBD [208]. Nonetheless, as pointed out in Section 2.7, physicians and others documenting in the EHRs commonly note typical symptoms or exposures and do not provide a comprehensive assessment of potential exposures (the ‘exposome’); this assessment is therefore subject to a broad range of biases, and the results must be interpreted carefully with a large measure of skepticism (i.e., ‘garbage in, garbage out’).

Medical informatics can also inform procedural quality standards and can create predictive risk severity models for different gastrointestinal pathologies. Mehrotra et al. used natural language processing to analyze a total of 24,157 free-text colonoscopy reports to determine provider performance based on different quality measures. Interestingly, they found a high variation of colonoscopy performances in an academic hospital, highlighting the need for better methods to assess the quality of colonoscopy procedures and consequently improve institutional quality [209]. Additionally, machine learning can be leveraged to create predictive scoring models for complex syndromes currently lacking a well-rounded clinical risk scoring system, as is the case for acute pancreatitis and upper gastrointestinal bleeding. By examining 15 markers routinely assessed in the blood from 300 patients with acute pancreatitis, Jin et al. demonstrated the machine learning model of multilayer perception—the artificial neural network reliably predicted disease severity and informed physicians in early management decisions [210]. Similarly, a machine learning model was validated and deemed superior to existing predictive risk scores in identifying the need for hospital-based interventions in very low-risk patients with upper gastrointestinal bleeding [211]. All in all, medical informatics clearly allow for the improvement and integration of clinical with multi-omics data, paving the way for the creation of GI-based large cohort studies that can improve clinical care.

### 3.8. Imaging Data 

As large collections of medical imaging data become available, there is much interest in the applications of machine learning to the development of computer-assisted diagnostic tools. The computer-assisted detection of abnormalities in screening colonoscopy or endoscopy has the potential to impact the practice of gastroenterology, given the wide range of adenoma detection rates dependent on endoscopist and situational factors [212,213]. In an early report of real-time, video-based automated polyp detection during colonoscopy published in 2016, Tajbakhsh et al. used a machine learning system to detect polyps based on their shape and by using visual context to filter out non-polyp background structures in the images, doing so with a latency of 0.3 s, a sensitivity ranging from 48% to 88%, and 0.05 to 0.1 false positives per frame of video [214]. More recently, Klare et al. conducted a clinical trial to determine the feasibility of real-time computer-assisted polyp detection during 55 screening colonoscopies and reported no adverse clinical events. In that trial, the detection rates of the polyps and adenomas by the computer-assisted system were approximately 51% and 29%, respectively, compared to 56% and 31% by routine colonoscopy. However, the study was limited by poor detection of small and flat polyps due to the system’s training being primarily on images of large polyps [215].

The literature on real-time, computer-assisted polyp detection in colonoscopy has flourished. Wittenberg and Raithel propose two main driving forces: the growing number of publicly available colonoscopy image and video datasets and the advent of “deep learning” capabilities by artificial neural networks that allow them to be trained using those datasets [216]. For example, Billah et al. used a linear support vector machine trained to detect polyps using over 14,000 images from screening colonoscopy videos, making it one of the largest studies of its kind at the time, achieving impressive accuracy, sensitivity, and specificity (each of approximately 99%) [217]. In general, deep neural networks trained using more colonoscopy sequence images appeared to achieve higher polyp detection rates, many with over 90% accuracy [218,219]. Systematic reviews of randomized controlled trials comparing computer-assisted polyp detection systems to the standard of care found that using computer-assisted detection significantly increased the detection rates for adenomas of diverse sizes and morphologies, and more adenomas and polyps were detected [220,221,222]. Although withdrawal time during computer-assisted colonoscopies is prolonged compared to routine colonoscopy, this difference appears to be due to the time spent on mucosal biopsies [222]. Such robust evidence supporting the use of these approaches has resulted in their rapid commercialization and adoption by many clinical practices.

## 4. Multi-Omics Data Integration 

In contrast to descriptive, single data-type approaches, integrating multi-omics data may uncover more detailed mechanistic information that can be validated with wet laboratory experiments (Figure 1). This is especially important when studying complex diseases where a singular etiology may not be responsible, and a goal is to apply the findings to precision medicine. IBD offers a quintessential, oft-studied example of how this can apply to a prevalent digestive disease. Several studies applied transcriptomics, metagenomics, and metabolomics, in the context of genetic variants associated with IBD [223,224,225,226,227], and identified potential therapeutic targets. Non-genetic associations between microbes, metabolites, and transcriptomics [228,229,230] uncovered promising diagnostic and therapeutic targets. Multi-omics approaches are an important aid for diseases involving multiple organ systems. For example, rat models of non-bacterial prostatitis revealed microbial changes associated with transcriptome and methylome changes at a single timepoint that indicated a potential role for intestinal immunity and inflammation [231]. Another example is diabetic kidney disease (DKD), where fecal microbes and metabolites correlated with increased levels of specific serum metabolites are associated with more rapid disease progression [232].

To ensure multi-omic studies are appropriately designed and conducted, multiple considerations must be weighed, most importantly the data type and sample size. Disease subtyping may have different requirements than those of biomarker identification [233], and these may differ based on the sample type characterized. Sample collection methods can impact sample size—for example, within the microbiome realm, swabs will have fewer human cells than mucosal biopsies, impacting the requirements to attain similar sequencing depth; increased depth permits smaller sample sizes. Sample collection for microbiome analysis may also influence results—microbial profiles from mucosa-based samples differ from those in stool [10,234], particularly in disease [10,235], and metabolome profiles may differ based on stool vs. serum assessment [236]. Likewise, based on the time of collection and other features, the results from rectal swabs differ from those of stool samples [237]. While these considerations apply to single-data type studies, they are particularly important for multi-omic studies, where the relationships are complex and often non-linear [238,239].

Several methods exist to integrate multi-omics data and the sample size, data type, and other study characteristics which are considerations when deciding which tools to use [233,240]. Most straightforward are simple correlation analyses, while more complex distance metrics and associated ordination methodologies are also commonly used. Network analyses take associations a step further and permit the linkage of whole pathways and groups of elements (microbes, genes, etc.), as opposed to identifying relationships at the individual level. This allows for deeper characterization, which is more likely to result in pathway identification [241,242]. Various mathematical methods exist for implementing these analyses, and each has strengths and biases. Matrix factorization methods (e.g., k-means clustering) are increasingly popular [243], with heavy reliance on artificial intelligence to eke out complex relationships; however, large sample sizes are required for this to be an effective tool [244,245]. Other clustering methods, such as hierarchical clustering, are often employed as well, particularly for taxonomic data, though other datatypes such as metabolome data are amenable to this analysis as well [246]. As described by Subramanian et al. [233], many tools have been designed for multi-omic integration such as an integrated multi-omic pipeline out of MIT [247], and others will likely be developed and refined as datasets become increasingly complex.

## 5. Challenges to Employing Big Data Research in GI

While the generation of large datasets for gastroenterology and hepatobiliary research increases, several hurdles must still be overcome for translational relevance. Broadly, these include the appropriate study design, analysis, and integration of data and the production of technologies that can aid in data generation, particularly at the multi-omic level. In the following sections, we provide suggested approaches to overcoming the remaining challenges, and in Figure 2, we offer an example of how a thoughtful study design can help mitigate some of these issues.

### 5.1. Working with Sparse Datasets

While the mathematical and interpretive implications differ across data types, sparse data are a potential concern for nearly every data type discussed in this review. Many omics data are referred to as sparse datasets due to gaps in non-zero intervals of data, and these data require tailored statistical approaches to analysis, including normalizing data to relative abundance [248]. The compositional nature of the data requires methods of analysis such as sparse CCA for correlational [249,250] and PERMANOVA for multivariate [251] comparisons. However, sparsity is not limited to omics data and can occur in clinical data too, particularly when the descriptor variables are not well distributed across categories, or there are few occurrences of the outcome of interest [252]. Across all data types, bias often overestimates likelihood (such as risk or effect modification) [252], but steps can be taken to adjust for this—the most straightforward is to simplify a model and avoid overfitting or to utilize conditional logistic regression when appropriate [253,254]. Additionally, penalized likelihood estimators can adjust for bias that may be difficult to avoid otherwise [255,256,257], though lower convergence rates may pose a problem [258]. Furthermore, a sparse data bias can be propagated by combinatory methods used for meta-analysis [259], but adjustments such as continuity correction can be applied to overcome these biases [260,261]. 

### 5.2. Accounting for Missing Data

Missing data pose a separate challenge—most statistical methodologies require complete datasets, and large datasets are often missing data. Both omics and clinical data commonly deal with this by filtration of low data variables from the dataset and missing value imputation [262]; filtration is straightforward, whereas imputation can be more difficult. Some datasets, such as microbiome data, rely so heavily on relative abundance normalization that zero-based imputation for microbes that are not expected to be present or abundant is reasonable. This is not the case for metabolomics and other datasets, where missing data can imply immeasurability rather than absence. Difficult to measure factors include volatile compounds, which are thought to play a strong role in the mechanism underlying many digestive disorders and are often missing from intestinal metabolomic data due to rapid degradation [11,12]. Others include compounds with a wide range of endogenous availability, where either side of the range may not be appropriately measured (i.e., the levels are too high or too low and a blank value is returned) [263]. Local imputation methods, such as k-nearest neighbors (KNN) and regression-based algorithms [262], as well as global clustering-based approaches [264], are common, and multiple R and Python packages can provide assistance [265,266]. Using data-appropriate algorithms is key to obtaining useful, reliable results [267], and care must be taken to consider which approaches are suitable, not only for a given data type, but also for specific datasets. 

### 5.3. The Role of Technological Advances 

Over the last decade, sequencing and other data generation technologies have rapidly improved, thereby transforming the omics and big data landscape. These developments have made the generation of large swaths of data feasible, allowing more data to be analyzed in a high-throughput fashion. To address gaps in the quality of the data produced, the analytical capability can be improved with technological advances, including improvements in the affordability of higher-throughput methods, multi-omics data generation methods, and portable devices permitting onsite data acquisition and analysis. Balancing the cost of data generation has always been of concern, particularly with high-throughput technologies [268]. As technology advances, a cost-prohibitive barrier often accompanies it. This is, in large part, to recover the costs of development. However, as technologies progress from being considered state-of-the-art to routine and widespread, they tend to become more affordable. The costs of higher-throughput data with increased depth are increasingly affordable, allowing choices such as metagenomic over 16S rRNA amplicon sequencing to become more common in microbiome analysis. Still, the importance of funding research that proposes newer, more innovative, and more expensive techniques cannot be understated. A recent “Request for Applications” (RFA) from the National Human Genome Research Institute (NHGRI; RFA-HG-22-008) seeks to fund the exploration of some of these questions more generally, while other RFAs are targeted to specific areas (e.g., RFA-AI-22-038 and multi-omics in HIV treatment and vaccination). Although academic biomedical research relies heavily on NIH support, industry funding provides an alternative revenue source, and several big biotechnology companies have shifted their funding towards projects exploring multi-omics as a means of diagnostic and therapeutic target identification and characterization. This can be seen both in the direct funding opportunities [269,270] and in the development of analysis platforms [271] and masterclasses for multi-omics analysis [272], geared towards encouraging multi-omics research in the broader bio-medical community.

Noise introduced by sample-site variation represents another barrier. Traditionally, multiple individual samples were used for varying types of data generation, and even when one sample was split into multiple components, different portions of the sample were used to generate a specific data type. When different types of data are pooled in multi-omics analysis, the use of different sampling locations can be a confounder due to differences in immediately adjacent tissues or sample sites such as blood. A well-known example in GI research is the datasets based on tissue obtained from liver biopsy; differences in liver biopsy histopathology not due to interobserver variation are common across multiple types of liver disease, and especially in cirrhosis [273,274,275]. Biopsies taken from different parts of the same cirrhotic liver may reveal substantial differences in histopathology; they can also display different transcriptional, epigenomic, metabolomic, and proteomic profiles. Another example is single vs. multi-site blood culture, where sampling methods can affect factors such as the infection positivity rate [276]. Thus, being able to utilize the same sample site for multiple types of data analysis is key to removing site variation-related noise and improving data resolution and statistical power. 

Fortunately, the technologies under development will permit the collection and generation of multiple types of data at once from a single sample. Visium spatial gene expression analyzers utilize hybridization for RNA and protein co-detection, while their single-cell technologies permit simultaneous cell-surface protein analysis [277]. NanoString offers similar services with CosMx and GeoMx spatial molecular imager technologies [278]. These machines, which are compatible with several tissue types, permit flexibility in sample choice as opposed to requiring fresh samples preserved with specific protocols. However, the development of more technologies that permit co-analysis of multiple data types is needed, and this poses a hurdle that, once overcome, will revolutionize the omics world. 

Lastly, sample processing techniques strongly impact data output, and delayed processing can substantially influence the data types prone to rapid fluctuations, including RNA-based, metabolite, protein, and even microbial data. RNA is prone to rapid degradation, as are certain volatile metabolic compounds, and while preservatives are frequently used, they may also prevent the use of techniques downstream. Portable devices, such as the MinION Oxford Nanopore Sequencer, permit the immediate analysis of data, thus removing the need for preservation [279] and allowing field sequencing. Swallowed “Smart Pills” can collect images or be used for sensory data collection in the GI tract [280]. Unfortunately, accuracy is often traded for convenience. Nanopore sequencers still have a high error rate [281], especially for RNA sequencing [282], making reference-free transcriptome analysis and strain detection very difficult. Furthermore, “Smart Pills” are limited in what and how many compounds they can sample, making it difficult to collect large volumes of data. Further development of portable devices will permit expanded capabilities and accuracy.

### 5.4. The Importance of Infrastructure and Multi-Center Initiatives

The context and potential application of findings using these methods and tools also warrant close consideration. The expansion of the infrastructure and computational ability inherent in electronic health records (EHRs) promises a new potential to use big data to augment the understanding of disease, but fragmented health care, numerous platforms, and lack of standardization of EHR data places significant limitations on this possibility. EHR platforms vary between services, clinical settings, institutions, regions, and countries, and are rarely linked in a functional manner that allows the effective collection and communication of information. Integrating data from multiple independent systems often requires labor-intensive and error-prone methods of manual data mining. In addition, bias can be introduced in subject inclusion and in the selection of relevant data points within these restrictions. Gastroenterologists assess disease activity and response to therapy in IBD using patient reports of symptoms, serum and fecal biochemical markers, endoscopic appearance, histopathology, and diagnostic imaging. The ability to analyze data points from distinct clinical, endoscopic, and radiologic systems is an essential precursor to employing methods such as machine learning to guide personalized and precise prognosis predictions and therapy selection [283]. The efforts to maximize cross-platform communication should be championed but will require notable investment in infrastructure and attention to security and privacy. 

In parallel to improving the accessibility and links between health information platforms, we must endeavor to standardize the quality of data entered into these systems. This encompasses the definition of relevant data points, the frequency at which they are collected, and the accuracy of their assessment. For example, in the aforementioned multifactorial assessment of IBD disease activity, visual images are interpreted by human operators to determine the degree of inflammation present, which is subject to significant interobserver variability. Computer-aided scoring systems can offer a more objective, reliable, and reproducible assessment of endoscopic, radiologic, and histopathologic images [284,285,286]. Adhering to the collection of information in a standardized manner is another step in the direction of optimizing data across the sprawling infrastructure of health information systems.

Critical to our understanding of the power of these methods of research is awareness of the potential shortcomings. It is established that >75% of included subjects in GWAS studies are of European ancestry [287]. Lack of diversity limits our understanding of the genetic basis of disease, interferes with the generalizability of the findings, and can exacerbate health inequities. Differences in linkage disequilibrium, specific gene effects within certain populations, and gene–environmental interactions greatly impact the ability to replicate identified genetic associations across diverse populations [288]. Genetic variation among populations can affect the efficacy and safety of drugs. Thiopurines have been used to treat IBD for decades and assessing thiopurine methyltransferase (TPMT) enzyme activity prior to initiation has long been standard practice. TPMT enzyme mutants with reduced activity are associated with life-threatening thiopurine-induced leukopenia in approximately 5% of patients of European descent [289]. However, despite lower TPMT variant frequency, a higher prevalence of thiopurine-induced leukopenia was observed in Asian populations [290,291]. A GWAS conducted in the Korean population identified a NUDT15 polymorphism associated with thiopurine-induced leukopenia. The effect size of this variant was greater than that for the TPMT variants in Koreans but is found in <1% of Caucasians [292]. This illustrates the importance of including subjects diverse in ethnicity and ancestry. In addition, dissimilarities in phenotype and cultural biases across global populations can impact the presentation and measurement of complex diseases. This necessitates diverse and multidisciplinary teams in the design and implementation of these studies, as well as the interpretation and application of their findings.

### 5.5. The Use (or Misuse) of Longitudinal Data 

The popularization of longitudinal sampling is an important next step in the big data and omics realm. Longitudinal sampling allows for the minimization of clinical heterogeneity, the characterization of sequential events, and the identification of dynamic relationships that cannot be observed in single-timepoint studies. Altogether, this fosters more insightful and mechanistically oriented and less generally descriptive research. The ability to meaningfully integrate multiple data types is key to tracking their relationships over time. Longitudinal data were used to better characterize GI [293] and non-GI [294,295] disorders and will become more widespread with increasing popularity. 

The appropriate study design and integration of longitudinal multi-omics data is required for proper interpretation [296]. Bodein et al. [297] pointed out that having access to longitudinal data does not guarantee insights that are immediately obvious from superficial analyses. By re-analyzing three previously published longitudinal datasets, they demonstrated how longitudinal data can be used to characterize dynamic and even potentially causal relationships (albeit requiring mechanistic validation). Furthermore, they recommend the use of pipelines that can discern functionality using multi-layer relationships from the data, as described in their “timeOmics” package [298]. As an organ, the luminal GI tract is especially amenable to the collection of multiple, longitudinal samples. Stool samples can be collected daily or as frequently as bowel movements, and samples can be used for multiple purposes; the collection of equivalent data in sera can also be obtained daily. When applicable, multiple colon biopsies can be taken on repeat colonoscopies with minimal to no increase in adverse events [299]. If designed with foresight and using questions that are well defined, studies of the GI tract are quite amenable to longitudinal multi-omic studies to answer the complex questions described above.

### 5.6. Technique and Pipeline Standardization in Big Data Analysis 

Lack of standardization and consistency in both technique and analytical pipelines is perhaps one of the most difficult hurdles to overcome in big data analysis. Non-standardized protocols often increase variance between study results based on assay type [300], sampling method and population [301], and research group experience [302]—even relatively minor differences in experimental conditions [303] that may be lab-specific [304], and even due to chance, both in the big data realm [305] and in broader biomedicine research [306]. For instance, intestinal microbiome and transcriptome composition assessed using stool samples can vary greatly depending on lysis methods [307]. Thus, due to inconsistent data collection and generation techniques and pipelines, reproducible results may not be attained [240,308] and, in fact, might play a larger role than currently appreciated; Xuan et al. reported that consistent results in data generation across multiple centers could be achieved when technical and analytical pipelines were heavily standardized [309]. The challenge, however, is that some degree of variability is needed to tailor experimental designs to the sample; for instance, while one set of kits may work best for samples stored in one buffer another kit may be compatible only with other buffers, and the buffers may have varying stabilities at room temperature (prior to freezing), and depending on study constraints, this may impact storage choices and thus kit choice. Another example is infection, which manifests differently in each animal model and even individual host and can make experimental reproducibility difficult [310]. Analytical pipeline choice is almost entirely dependent on dataset characteristics, study design, and other study specific factors, making it difficult, if not impossible, to obtain complete formal standardization. Nonetheless, some aspects can be more formally standardized than at present. As a field, GI and hepatology investigators can standardize several steps in the pipeline, including the following. 

The collection of metadata and clinical details, when specified, can control for confounders and the comparing of differences in study populations with a greater degree of precision [311]. For instance, always including a Bristol stool score when stool samples are used or consistently including information on the presence or absence of common disorders improves comparability between studies and the ability to control for confounding variables in meta-analyses.

The standardization of serum, tissue, and other sample collection protocols can greatly improve reproducibility. For example, the sampling location can greatly affect both the microbiome [312] and the metabolome [313], depending on the storage conditions and analytical methods (e.g., 16S rRNA sequencing vs. metagenomic sequencing), etc., and homogenization of the stool should always be performed prior to sample processing and data generation. This is particularly pertinent to GI research where the stool is commonly collected and assessed. 

For omics data, quality control of the data is fundamental to their reproducibility and trustworthiness [314]. Minimum standards must be more universally emphasized and enforced for publishing omics data.

The use of workflow management systems allows for the semi-standardization of analytical pipelines and decreases the major analytical variability that can drastically change interpretation [315,316,317]. These are useful both for omics [318] and clinical datasets [319].

The sample storage conditions and the technical and analytical processing steps should always be recorded in abundant detail to allow comparison between experiments. Minimizing the confounding introduced by other variables reduces dataset variability overall, thus augmenting the accuracy of the results. Large datasets or experiments with multiple replicates reduce noise while multicenter studies improve the ability to home in on true signals, for both clinical and omics data. The standardization of research pipelines can greatly improve the quality of big data in GI and hepatobiliary research [320] and is an important step in facilitating the widespread usability and reproducibility of omics and large dataset-based research.

### 5.7. Recognizing the Shifting Role of Clinical Trials 

The use of big data has the potential to revolutionize the design of clinical research. Although randomized controlled trials (RCT) are considered the gold standard and are able to control for non-randomized confounders, they are time- and manpower-intensive, costly, and have limited ability to detect rare or long-term effects [321]. In addition, they may not reflect the real-life effects of diseases or therapeutics and cannot be conducted in some circumstances due to ethical limitations [322]. The rigors and restrictions in recruiting subjects can limit generalizability and present challenges in studying populations that are vulnerable and/or underrepresented in clinical research. Studies using pre-existing swaths of data have the advantage of analyzing datasets that are easily and rapidly accessible, depend more upon computer software than manpower, and can take into account comorbidities and medical diversity that may impact outcomes [323]. The volume and variety of data enable the investigation of uncommon events over long periods of time and can include observations in diverse populations. Additionally, the capacity to simultaneously investigate a multitude of variables and conduct multiple sensitivity analyses allows for a more robust examination of the secondary endpoints and sub-groups that are often ignored in RCTs [324]. 

There are concerns regarding the residual or unmeasured confounders that can result in the inability to distinguish between association and causation [323]. In this context, considering the distribution of missing data and the potential for the misclassification of confounders or even variables of interest is important [323]. The inclusion of RCT datasets, the application of the Bradford Hill criteria, and the analysis of unstructured data within EHRs with natural language processing have been proposed as potential solutions [325]. The optimal approach to harnessing the advantages and minimizing the disadvantages of each approach to research may be to apply them sequentially. RCTs can confirm or validate hypotheses generated from the analysis of big data, and large clinical dataset analysis can explore RCT findings more deeply [324].

## 6. Conclusions and Future Directions 

A variety of holistic multi-omics approaches can be used to depict the variety of human and microbial genetic, transcriptomic, proteomic, and metabolomic events that underly physiological and pathophysiological phenomena. Thus, generating big data from multi-omic, multi-site studies has the extraordinary capability to enhance comprehensive investigations into normal and diseased GI and hepatobiliary function and, thereby, uncover important connections between these organ systems and neural, immune, and endocrine cells and gut microbial organisms. We provided specific examples to show how these approaches can be integrated to study the longitudinal effects of dietary, pharmacological, and other interventions. 

Although the power of these approaches to advancing our knowledge is unprecedented, the complexity of big data use and analysis lends itself to numerous potential pitfalls and the introduction of errors in study design, execution, and analysis that can compromise both the study results and their interpretation. Although we focus on the application, integration, and analysis of big data in gastroenterology and hepatobiliary research, these issues have widespread implications. In addition to expanding and improving the technology and computerization needed to collect, curate, and interpret the reams of multi-omics data with even greater precision, accuracy, and detail, future directions must include approaches that prevent mistakes and the misuse of such complex information. Such approaches include formalizing the standardization of big data derivation and analysis—even small changes in the study protocols and research and analytical designs can result in major differences in outcome. In this context, for example, we should encourage efforts to benchmark and improve the consistency of the approaches to performing single-cell genomic studies [326,327]. The expanded training of a large cadre of bioinformaticians adept in handling, storing, analyzing, and interpreting big data must be prioritized.

## Figures and Tables

**Figure 1 ijms-24-02458-f001:**
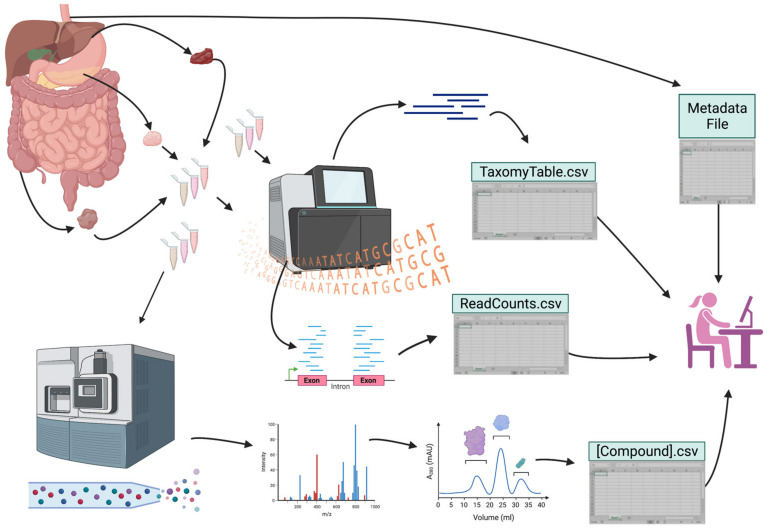
A pipeline for multi-omics data generation and analysis. Luminal GI, hepatobiliary, and pancreatic tissue can be sampled, homogenized, and used to generate multiple types of data from the same sample, such as DNA and RNA sequencing, as well as metabolomic and proteomic mass spectrometry-based and NMR-based data. These data can then be quality checked, cleaned, and processed into final datasets which can then be incorporated into a set of integrative analyses. Created with BioRender.com. novel hypotheses. Created with BioRender.com. D, day; Wk, week.

**Figure 2 ijms-24-02458-f002:**
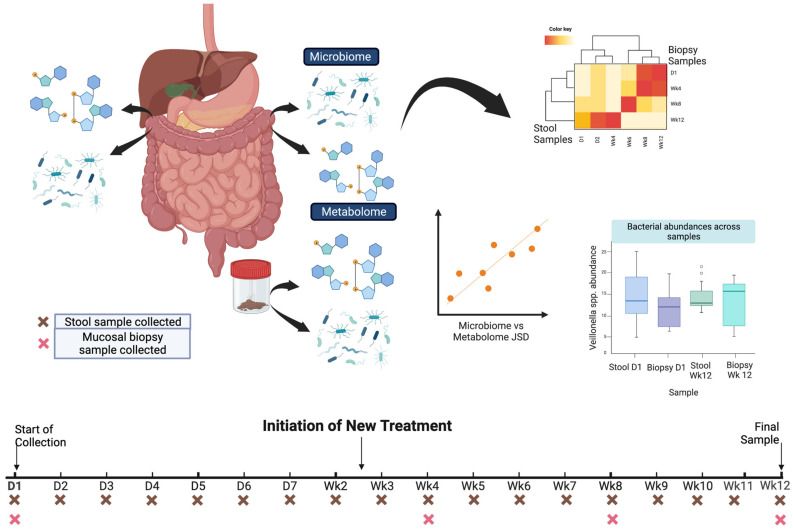
Example of a research design illustrating the benefits of using longitudinal multi-omics data in the context of evaluating changes in IBD treatment. Using stool and biopsy samples from multiple sites in the colon permits assessment of localized biosignatures that can be correlated for the development of diagnostics and therapeutics. Both pre- and post-treatment analyses facilitate the detection of biosignatures predicting therapeutic response. Using multi-omics data permits the inclusion of changes in the microbiome with genomic and metabolomic data—a holistic approach is also likely to generate this.

**Table 1 ijms-24-02458-t001:** Benefits and limitations to using particular datasets in GI and hepatobiliary research.

Datasets	Common Data Types	Benefits in GI and Hepatobiliary Research	Possible Limitations	Refs
Genetics/ Genomics	Whole genome, whole exome sequencing data	Many GI and hepatobiliary disorders have an under-characterized hereditary component	Most diseases are multifaceted—a large amount of data is needed to reveal true signals	[1]
Epigenetics/ Epigenomics	Most commonly methyl-seq, ChIP-seq, and ATAC-seq	These data may offer enhanced diagnostic utility, particularly in the context of GI and hepatobiliary disorders with complex genetic etiologies	Multiple factors such as the gut microbiome and diet greatly affect epigenetic regulation Detecting changes that can be universally tracked may be difficult without access to large amounts of data and clearly characterized subgroups Large number of cell types may render data interpretation difficult	[2]
Transcriptomics	mRNA-seq, total RNA-seq, targeted RNA-seq, scRNA-seq, and snRNA-seq	Metatranscriptomics offers insights into the transcriptional activity of intestinal microbes, whose presence does not always correlate with bacterial activity Digestive organs comprise complex cell types with specific biomarkers that make them excellent candidates for scRNA-seq analyses	Transcript expression does not always correlate with bacterial or human protein output; so, downstream validation must be performed to confirm findings The complex distribution of cell types makes it difficult to use or interpret bulk RNA-seq and other less costly methods for digestive organ research	[3,4,5]
Proteomics	NMR, integrated chromatography, and mass spectrometry data	Captures larger compounds than metabolomic analysis; this may be important in biomarker identification and validating transcriptional activity	Remains cost-prohibitive	[6]
Microbiome	16S rRNA amplicon sequencing, shotgun metagenomic sequencing	The intestinal microbiome contains the highest concentration of commensal bacteria residing on human tissue Large amounts of stool are relatively easy to collect with little human DNA present	Findings derived from stool may differ from those derived from mucosal biopsies (i.e., taken at the source) Association does not imply causation: bacterial profiles may change due to disease; findings must be carefully validated to attribute diseases to dysbiosis	[7,8,9,10]
Metabolomics	Liquid chromatography, gas chromatography, capillary electrophoresis, and ionic mobility spectrometry mass spectrometry, NMR data	Useful for gut volatile compounds, the metabolites thought to be most associated with disease Metabolomics combined with gut microbiome data can reveal mechanistic targets—particularly useful in study of GI and hepatobiliary disorders	The preservation of volatile compounds requires use of a buffer or immediate sample processing and may still not adequately capture their presence or abundance Host variation in endogenous compounds, particularly those interacting with bacterial pathways, may complicate development as a diagnostic tool	[11,12]
Medical informatics	EHR, questionnaire, patient interview data	Procedures such as colonoscopy and FibroScan for GI and hepatobiliary disorders, respectively, have quantifiable datapoints that can be combined with clinical and demographic data for retrospective and prospective research	Inconsistent techniques and data input and variation require rigorous coordination, quality assessments, and large cohorts to accurately capture differences	[13]
Imaging data	X-ray, CT, MRI, endoscopy, capsule imaging data	AI-based interpretation of upper, lower, and video capsule endoscopy can capture findings that may otherwise be overlooked	Subjective interpretation by the operator is still required (relatively minor concern since these can be standardized)	[14,15]

Refs, reference; ChIP, chromatin immunoprecipitation; NMR, nuclear magnetic resonance; EHR, electronic health record; CT, computerized axial tomography; MRI, magnetic resonance imaging; AI, artificial intelligence.

## Data Availability

No new data were created or analyzed in this study. Data sharing is not applicable to this article.

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
