# Peer review of "Big Data in Gastroenterology Research"

_ijms, 2023, doi:10.3390/ijms24032458_

Round 1

Reviewer 1 Report

Alizadeh et al. provide a very high level overview of omics in gastroenterology and hepatology.

The topic is of interest to broad audience.

The work is suitable for publication after minor revision.

Table 1 epigenetics – there is also the problem of confounding from cell type mixture proportions (not only transcriptomics)

Table 1 “FibroScan for GI and hepatobiliary disorders” – maybe only hepatic disorders? Fibroscan does not have large utility in the diseases of the intestine.

L130 why introduced “RNA-seq or RNA sequencing” if RNA-seq was already used earlier in the manuscript?

L131 “a” not needed at the end of the line

scRNA-seq – the problem is not only the cost but also geographical accessibility of centers/companies that offer scRNA-seq – preservation media can be used, but there are known problems and there is ongoing work on new preservation media that would allow transport – but this is not an important point

L222 “Commonly, third-party companies to generate the data” please correct (grammar/)

L294 in IBD it is more than 240 loci and even more were implicated by TWAS (then in line 296 “more than 100 loci”)

Line 397 – Could you please add just one example of such similarities between NAFLD and ALD?

Line 475 – It is not sufficient that a panel perfectly differentiates between patients with IBD and healthy subjects – it should optimally also be useful in differential diagnosis (e.g., IBD activity vs infection)

Line 475 also – What we actually do is – we characterize inflammation more precisely so that inflammatory bowel disease can be identified. However, it would be of great use if a biomarker was found that identifies IBD patients in remission. Inflammation is behind all our biomarkers. But the disease persists even with no active inflammation. This is where the problem of biomarker discovery meets the challenge of pinpointing the cause of IBD (and therapeutic targets) – they are all intertwined, as was already mentioned in the manscript. – no changes in text needed, just a comment

Line 557 “In FMT, are donated” – what is donated?

Line 585 “In healthy vs. IBD patients” or “In patients with IBD vs healthy controls” ?

Line 628 I think there is a lot of bias in data from EHR – physicians more often note typical symptoms or exposures. Please refer to the exposome in this part, e.g. “However, EHR do not provide comprehensive assessment of, e.g., the exposome, and are susceptible to a broad range of biases.”

Line 638 “15 blood routine markers” or “15 markers routinely assessed in the blood”?

Line 656 – the examples and reference no. 214 are from 6 years ago. Given the extremely rapid development in the field there are surely much better examples that demonstrate higher precision of AI in endoscopy. I suggest not to report the diagnostic values which are no longer relevant.

Line 691-693 I am not convinced by the example of prostate-gut axis in rat. Could a study in human patients be provided as an example of such integration instead (or additionally)?

Line 719 – Not only large datasets are required, but also the integration of rich sources of information (prior knowledge). Moreover, AI models are often impossible to understand – they may work, but it is not easy to make them provide new knowledge of mechanisms (especially with regard to neural networks).

Line 733 section on sparse data – I would not agree that most omics data are sparse and it is not very frequent that clinical data are sparse. A sparse matrix of n x p contains approximately n or p elements – this is really a sparse matrix. In my view, transcriptomic or epigenomic data are not sparse, they are dense. Maybe scRNA-seq data could be considered sparse (?). However, such data are filtered before further analyses. If I misunderstand something or lack knowledge please help by explaining.

Line 774 – gaps between what?

Line 786 – “RFA” need to be introduced before being used

Line 875 – “However, a despite lower” – “a” needs to be removed?

Author Response

Responses to Reviewer 1

Alizadeh et al. provide a very high level overview of omics in gastroenterology and hepatology. The topic is of interest to broad audience. The work is suitable for publication after minor revision. Response: We thank the reviewer for kind comments and for providing numerous comments and suggestions that helped us improve the quality of our manuscript.

Table 1 epigenetics – there is also the problem of confounding from cell type mixture proportions (not only transcriptomics). Response: In response to the reviewer’s concern, we acknowledged this point in the revised Table 1.

Table 1 “FibroScan for GI and hepatobiliary disorders” – maybe only hepatic disorders? Fibroscan does not have large utility in the diseases of the intestine. Response: In response to the reviewer’s concern, in the revised Table 1, we added the word ‘respectively’ to clarify this point.

L130 why introduced “RNA-seq or RNA sequencing” if RNA-seq was already used earlier in the manuscript? Response: In response to the reviewer’s concern, in the revised manuscript, we moved the preceding sentence to a more appropriate location further in the text (L148) so that this sentence is the initial introduction of RNA sequencing.

L131 “a” not needed at the end of the line. Response: Thank you, we corrected this typographical error.

scRNA-seq – the problem is not only the cost but also geographical accessibility of centers/companies that offer scRNA-seq – preservation media can be used, but there are known problems and there is ongoing work on new preservation media that would allow transport – but this is not an important point. Response: In response to the reviewer’s concern, we added a line to the revised manuscript acknowledging this.

L222 “Commonly, third-party companies to generate the data” please correct (grammar/). Response: The word ‘to’ was removed from the revised manuscript.

L294 in IBD it is more than 240 loci and even more were implicated by TWAS (then in line 296 “more than 100 loci”). Response: We agree. In the revised manuscript we corrected the text to reflect the many new IBD-associated genetic loci identified after publication of the work by Jostins et al. in 2012.

Line 397 – Could you please add just one example of such similarities between NAFLD and ALD? Response: As suggested, in the revised text, we added the example of COL genes involved in the regulation of hepatic fibrosis.

Line 475 – It is not sufficient that a panel perfectly differentiates between patients with IBD and healthy subjects – it should optimally also be useful in differential diagnosis (e.g., IBD activity vs infection). Response: In response to the reviewer’s concern, in the revised manuscript, we added text to clarify that biomarkers are not only important diagnostically, but also for excluding active disease. We mentioned CMV colitis as such a potential confounder.

Line 475 also – What we actually do is – we characterize inflammation more precisely so that inflammatory bowel disease can be identified. However, it would be of great use if a biomarker was found that identifies IBD patients in remission. Inflammation is behind all our biomarkers. But the disease persists even with no active inflammation. This is where the problem of biomarker discovery meets the challenge of pinpointing the cause of IBD (and therapeutic targets) – they are all intertwined, as was already mentioned in the manuscript. – no changes in text needed, just a comment. Response: This is an excellent point. Being able to differentiate between active IBD and quiescent IBD in a patient with another intestinal disease (e.g., CMV colitis) would be ideal; we clarified this in the revised text.

Line 557 “In FMT, are donated” – what is donated? Response: For clarification, in the revised manuscript, we added the word ‘microbes.’

Line 585 “In healthy vs. IBD patients” or “In patients with IBD vs healthy controls” ? Response: We revised the text to reflect the reviewer’s helpful suggestion.

Line 628 I think there is a lot of bias in data from EHR – physicians more often note typical symptoms or exposures. Please refer to the exposome in this part, e.g. “However, EHR do not provide comprehensive assessment of, e.g., the exposome, and are susceptible to a broad range of biases.” Response: The reviewer makes an excellent point. In the revised manuscript, we added text to both section 2.7 and the section on the EHR to highlight this concern. We thank the reviewer for suggesting the appropriate wording.

Line 638 “15 blood routine markers” or “15 markers routinely assessed in the blood”? Response: We revised the text to incorporate the reviewer’s suggestion.

Line 656 – the examples and reference no. 214 are from 6 years ago. Given the extremely rapid development in the field there are surely much better examples that demonstrate higher precision of AI in endoscopy. I suggest not to report the diagnostic values which are no longer relevant. Response: In the revised manuscript, we clarified that those results were published in 2016 and provide the results from more recent studies immediately follow this reference.

Line 691-693 I am not convinced by the example of prostate-gut axis in rat. Could a study in human patients be provided as an example of such integration instead (or additionally)? Response: In response to the reviewer’s concern, in the revised manuscript, we added a human example regarding the progression of diabetic kidney disease.

Line 719 – Not only large datasets are required, but also the integration of rich sources of information (prior knowledge). Moreover, AI models are often impossible to understand – they may work, but it is not easy to make them provide new knowledge of mechanisms (especially with regard to neural networks). Response: We agree -- for these to be truly effective, the integration of established knowledge is vital. Nonetheless, we are sure the reviewer will agree with us that advances in this area are proceeding at a breathtaking pace.

Line 733 section on sparse data – I would not agree that most omics data are sparse and it is not very frequent that clinical data are sparse. A sparse matrix of n x p contains approximately n or p elements – this is really a sparse matrix. In my view, transcriptomic or epigenomic data are not sparse, they are dense. Maybe scRNA-seq data could be considered sparse (?). However, such data are filtered before further analyses. If I misunderstand something or lack knowledge please help by explaining. Response: The definition of sparsity being min(n,p) elements present applies to the entirety of the matrix, however, data can have sparse columns or variables without the entirety of the matrix being sparse and, in fact, this is often the case.  If those columns are of interest in analysis, then our statements still apply. Further, as the reviewer mentions, filtering frequently occurs and alters the distribution of the dataset, thus potentially introducing bias – the property of sparsity still impacts the data, despite methods commonly used to handle this. While sparsity in a matrix of clinical data is unlikely, sparsity in a column of clinical data is not uncommon. This is particularly relevant when exploring rare or uncommon diseases, or procedures, in the context of large datasets. Nevertheless, we admit to being too general in stating “notable issue for every data type” and “omics data are referred to as sparse”, and edited the text to read: “Many omics data…”  and “potential concern for nearly every data type” (line 762-3) to reflect that not all omics datasets fall into this category.

Line 774 – gaps between what? Response: In response to the reviewer’s concern, in the revised manuscript, we clarified this portion of the text.

Line 786 – “RFA” need to be introduced before being used. Response: In response to the reviewer’s concern, in the revised manuscript, we defined RFA at its first use.

Line 875 – “However, a despite lower” – “a” needs to be removed? Response: Thank you, this typographical error was deleted from the revised text.

Reviewer 2 Report

The manuscript presented here titled "Big Data in Gastroenterology Research" is a necessary and thorough review about -Omics technologies and Diagnostic techniques used in research in biomedicine, specifically in Gastroenterology. I have find specially interesting the section 5 about challenges to undertake to improve the future management and production of Big Data. Also, I'm specially agree with the concluding remark of authors about the importance of the training of bioinformaticians and their key role for the data integration.

However, the current manuscript still have some issues making it not ready for its publication. In general, manuscript is too long and some sections should be shortened, using more concise language. Authors should avoid repetition of concepts between sections. Also, the use of so many references is remakable, but there are too many specific mentions in the text to specific examples of research. Some of these mentions just make the text longer where the point of the text was already made and exemplified in that section.

Section 2 could be directly removed. Authors are already explain thoroughly each dataset type on section 3, so there is no need to explain them previously on section 2. If authors consider that something is missing without the section 2, they could add some details on the introduciton or section 3.

Table 1 could be improved, adding a specific brief description about the origin of each dataset and example of application. Also, text on each cell could be reduced and shortened to be more straighforward for a quick reading.

Something that specifically bothers me is the treatment made in general in the manuscript to the Microbiome. Firstly, all -Omics techniques mentioned in this work are applicable specifically to the microbiome, not only 16S and metagenomics, but also proteomics (metaproteomics). Besides, regarding metabolomics, host-gut microbiome metabolomic profile is intimately related with microbiome. There are many examples in the literature of their application, specially to the gut microbiome. However, in this manuscript microbiome is shown like a small part of the overall gastroenterology research when it is a whole area of research itself. 

Figure 1 is good, but I miss an additional mention to proteomics analysis.

Figure 2 is a bit poor, if authors just want to illustrate how longitudinal research should be perform they could just add a similar time line as there is  at the bottom of figure 2 to the figure 1. However, I miss a figure illustrating an example of research using all mentioned datasets and examples of the information taking from them after integration.

Sections mentioning Medical Informatics and Data Integration could be part of the same main section about Bioinformatics in Biomedicine. Bioinformatics includes all described in both sections, processing of data, text minning, statistics, standarization, correlation of datasets, network analysis, etc.

Author Response

The manuscript presented here titled "Big Data in Gastroenterology Research" is a necessary and thorough review about -Omics technologies and Diagnostic techniques used in research in biomedicine, specifically in Gastroenterology. I have find specially interesting the section 5 about challenges to undertake to improve the future management and production of Big Data. Also, I'm specially agree with the concluding remark of authors about the importance of the training of bioinformaticians and their key role for the data integration. Response: We thank the reviewer for kind comments.

However, the current manuscript still have some issues making it not ready for its publication. In general, manuscript is too long and some sections should be shortened, using more concise language. Authors should avoid repetition of concepts between sections. Also, the use of so many references is remakable, but there are too many specific mentions in the text to specific examples of research. Some of these mentions just make the text longer where the point of the text was already made and exemplified in that section. Response: As requested by the reviewer, we revised the manuscript to avoid redundancy and to make our comments more concise.

Section 2 could be directly removed. Authors are already explain thoroughly each dataset type on section 3, so there is no need to explain them previously on section 2. If authors consider that something is missing without the section 2, they could add some details on the introduciton or section 3. Response: We respectfully disagree. We believe Section 2 provides important context for the entire review that does not fit well within the Introduction.

Table 1 could be improved, adding a specific brief description about the origin of each dataset and example of application. Also, text on each cell could be reduced and shortened to be more straightforward for a quick reading. Response: To avoid making the text in each cell overly long and cumbersome, we added a new column to Table 1 with a short descriptor about the origin of each datatype. Many examples of applications are provided throughout the text - this was not the goal of Table 1. We also believe it would be counterproductive to add these to what is already a ‘busy’ table.

Something that specifically bothers me is the treatment made in general in the manuscript to the Microbiome. Firstly, all -Omics techniques mentioned in this work are applicable specifically to the microbiome, not only 16S and metagenomics, but also proteomics (metaproteomics). Besides, regarding metabolomics, host-gut microbiome metabolomic profile is intimately related with microbiome. There are many examples in the literature of their application, specially to the gut microbiome. However, in this manuscript microbiome is shown like a small part of the overall gastroenterology research when it is a whole area of research itself. Response: To clarify, our goal was to review different types of omics and large datasets in the context of GI and hepatology research – not to focus on microbiome research in GI, which is itself an entire exhaustive field. Further, whereas “all omics techniques mentioned are applicable specifically to the microbiome”, they are also applicable to the human organism. So, it is not clear why we would only discuss human genomic sequencing as its own category but not microbial? The phrase ‘microbiome data’ is commonly used by those in the field to imply the study of the abundance or composition of microbial communities. Studying the abundance of various species is not the same as examining what they produce (metabolomics, proteomics, etc.). Just as microbes produce metabolites, so do humans. Thus, including metabolomics as part of the microbiome section does not seem appropriate. Instead, we separated microbial sequencing and human/eukaryotic sequencing from the other data types and provided extensive discussion of the common uses of metabolomics data within the context of microbiome research, just as we discussed potential applications of studying human metabolomics. We discussed the importance of integrating these data at several places in the manuscript.

Figure 1 is good, but I miss an additional mention to proteomics analysis. Response: Thank you. In response to this concern, in the revised manuscript, we modified the figure caption to include proteomics.

Figure 2 is a bit poor, if authors just want to illustrate how longitudinal research should be perform they could just add a similar time line as there is at the bottom of figure 2 to the figure 1. However, I miss a figure illustrating an example of research using all mentioned datasets and examples of the information taking from them after integration. Response: We respectfully disagree with both comments. We believe Figures 1 and 2 add important complementary information to the manuscript, particularly by clarifying how such data can be integrated to provide a comprehensive, holistic view of otherwise discrete complex processes. Figure 2 provides an important example of how this approach can be applied to a specific gastrointestinal disorder, IBD. As the caption for Figure 2 indicates, this illustrates ‘an example of research using all mentioned datasets and examples of the information taking from them after integration’; indeed, this is specifically what the reviewer requests.

Sections mentioning Medical Informatics and Data Integration could be part of the same main section about Bioinformatics in Biomedicine. Bioinformatics includes all described in both sections, processing of data, text minning, statistics, standarization, correlation of datasets, network analysis, etc. Response: Medical informatics and data integration are discrete concepts. Medical informatics refers to the study of patient clinical data, including laboratory results. Data integration refers to using data of different types, including clinical data, and incorporating them. Bioinformatics refers to the development of methodologies and tools to study these data but does not imply that multiple data types are being integrated – for instance, one could choose to only look at RNAseq data in a bioinformatic analysis. Correlation is a statistical concept that examines the association between data – again, this can be across different data types (in the case of integrating data), or in a singular data type (e.g., using only RNA or medical informatics data). Similar sentiments are relevant to the other “keywords” mentioned. Nonetheless, we agree that the framing and introduction of section 2.7 was insufficiently specific to medical informatics and added text to discuss these methods and models as they apply to medical informatics, as well as some potential pitfalls. We hope the edits in the revised text clarified the differences between these sections and emphasized why we wish to keep them separate.

Round 2

Reviewer 2 Report

Authors disagree in the half of my coments, what is a bit disappointing. However, they have made a great efford improving the manuscript with the comments they found useful and defending their conceptualization of the manuscript.

Although I would use a different structure for the text, that doesn't mean the manuscript isn't ready to be published. I find its content a key piece of information about the current status of the use of -omics technologies, bioinformatics and the processing and integration of the huge bulk of data produced by modern technologies in biomedical research.